

# Clinical impact of lenvatinib in patients with unresectable hepatocellular carcinoma who received sorafenib

Yen-Yang Chen[1], Chih-Chi Wang[2], Yueh-Wei Liu[2], Wei-Feng Li[2] and Yen-Hao Chen[1,3,4]

[1] Department of Hematology-Oncology, Kaohsiung Chang Gung Memorial Hospital and Chang Gung University College of Medicine, Kaohsiung, Taiwan
[2] Division of General Surgery, Department of Surgery, Kaohsiung Chang Gung Memorial Hospital and Chang Gung University College of Medicine, Kaohsiung, Taiwan
[3] School of Medicine, Chung Shan Medical University, Taichung, Taiwan
[4] Department of Nursing, Meiho University, Pingtung, Taiwan

## ABSTRACT

**Background.** Lenvatinib has been approved for use in the systemic treatment for unresectable hepatocellular carcinoma (HCC). This study aimed to investigate the efficacy and safety of lenvatinib in patients with unresectable HCC who received sorafenib.

**Methods.** A total of 40 patients who received lenvatinib after sorafenib were retrospectively identified: as second line in 20 patients, third line in 10 patients, and fourth line and later lines in 10 patients. The treatment response to lenvatinib was determined in accordance with the guidelines of the modified Response Evaluation Criteria in Solid Tumors (mRECIST) every 2–3 months after commencement of lenvatinib.

**Results.** Median progression-free survival (PFS) and median overall survival (OS) of the whole population were 3.3 and 9.8 months, respectively. The objective response rate was 27.5%. Univariate and multivariate analyses showed that alpha-fetoprotein level >400 ng/mL was an independent prognostic factor of worse PFS and OS. The clinical outcomes of lenvatinib therapy as second-line, third-line, or fourth line and later line treatment were similar, and previous response to sorafenib could predict the response to subsequent lenvatinib. Most adverse events were grades 1–2, and the majority of patients tolerated the side effects. Our study confirms the efficacy and safety of lenvatinib as second-line and later line treatment for patients with unresectable HCC who received sorafenib in clinical practice.

## INTRODUCTION

Hepatocellular carcinoma (HCC) is one of the most common aggressive malignancies worldwide and is the second leading cause of cancer-related deaths in Taiwan (*Ministry of Health and Welfare of Taiwan, 2015*). The well-known risk factors for HCC include hepatitis B, hepatitis C, and alcoholic and nonalcoholic steatohepatitis. Despite surveillance programs in high-risk patients, some patients with HCC have advanced status when

Corresponding author
Yen-Hao Chen,
alex2999@cgmh.org.tw,
alex8701125@gmail.com

diagnosed, including multicentric intrahepatic spread, portal vein thrombosis (PVT), huge tumor burden, and distant metastasis, contributing to poor prognosis (*Bertuccio et al., 2017*; *Bruix, Reig & Sherman, 2016*; *Cabibbo et al., 2012*).

Sorafenib is a small oral tyrosine and serine/threonine kinase inhibitor (TKI) that has been proven as the first-line systemic treatment for unresectable HCC (*Cheng et al., 2009*; *Llovet et al., 2008*). Before 2018, sorafenib therapy is the only recommended strategy that can prolong overall survival; apart from this, no other approved targeted therapy in the past 10 years has been available as the first-line treatment for unresectable HCC (*Cainap et al., 2015*; *Cheng et al., 2013*; *Johnson et al., 2013*). Recently, lenvatinib, a newly developed TKI targeting vascular endothelial, fibroblast, and platelet-derived growth factor receptors, has become available as the first-line treatment. The REFLECT trial showed that lenvatinib led to significantly better progression-free survival (PFS), time to progression (TTP), and objective response rate (ORR) than sorafenib and was non-inferior to sorafenib in terms of overall survival (OS) (*Kudo et al., 2018*). Since then, an increasing number of patients have been receiving lenvatinib therapy as the first-line treatment in many countries, especially in Japan and Taiwan.

In the SELECT trial, a study that focused on the use of lenvatinib in radioiodine-refractory thyroid cancer, there was no significant difference in survival between patients who received and did not receive sorafenib (*Schlumberger et al., 2015*). This phenomenon was also observed in patients with HCC. Recently, only few studies have showed no difference in survival outcomes between patients who received targeted therapy and treatment-naïve patients (*Hiraoka et al., 2019a*; *Hiraoka et al., 2019b*; *Hiraoka et al., 2019c*). However, to the best of our knowledge, information on the clinical impact of lenvatinib, such as prognostic factors, adverse events, and correlation between sorafenib and lenvatinib response, in patients with HCC who received sorafenib is limited in literature. The present study aimed to investigate the clinical impact and safety of lenvatinib in patients with HCC who received sorafenib, including prognostic factors of PFS and OS, survival comparison between different treatment line, presentation of side effects, and correlation of response of lenvatinib and prior sorafenib.

## MATERIALS & METHODS

### Patient selection

Between September 2018 and December 2019, the records of patients with unresectable HCC who underwent treatment at the Kaohsiung Chang Gung Memorial Hospital were reviewed retrospectively. First, only patients who received sorafenib followed by lenvatinib were enrolled. Second, the included patients must have had at least one measurable target lesion for the evaluation of treatment response. Third, the eligible patients had an Eastern Cooperative Oncology Group Performance Status score of 0–1 and controlled hypertension. We excluded patients with a history of second primary malignancy or concurrent hepatocholangiocarcinoma. Finally, 40 patients who had available medical records and follow-up visits were identified.

### Diagnosis and staging of HCC

The diagnosis of each patient with HCC was made according to pathology and dynamic computed tomography (CT) or magnetic resonance imaging (MRI) of the liver in high risk patients (HBV or HCV related liver cirrhosis) (*Marrero et al., 2018*; *Omata et al., 2017*). Staging was determined according to the Barcelona Clinic Liver Cancer (BCLC) staging classification at the time of lenvatinib treatment initiation (*Llovet, Bru & Bruix, 1999*). Alpha-fetoprotein (AFP) was measured before commencement of lenvatinib for each patient.

### Lenvatinib treatment and assessment of adverse events

Lenvatinib was orally administered at a dose of 10 mg and 8 mg once daily for patients with body weight $\geq$ 60 kg and <60 kg, respectively. Adverse events (AEs) were evaluated according to the National Cancer Institute Common Terminology Criteria for Adverse Events version 4.0, and the worst grade for each AE was recorded (*National Cancer Institute, 2009*).

### Evaluation of treatment response

Each patient must have had at least one measurable target lesion and enhanced CT or MRI of the liver to be evaluated for treatment response to lenvatinib in accordance with the guidelines of the modified Response Evaluation Criteria in Solid Tumors (mRECIST) (*Lencioni & Llovet, 2010*) every 2–3 months after commencement of lenvatinib. Assessment was performed independently by two radiologists blinded to any information about the patients' clinicopathologic features or prognosis.

### Statistical analysis

Statistical analyses were performed using SPSS 19 software (IBM, Armonk, NY, USA). Statistical analyses of group differences were performed using the Chi-square test with Bonferroni correction for categorical variables. PFS was defined as the time from the initiation of lenvatinib treatment to disease progression or death from any cause, and OS was calculated from the date of lenvatinib treatment initiation to death or last living contact.

Kaplan–Meier method was used to estimate PFS and OS, and the log-rank test was performed to evaluate the differences between groups for univariate analysis. The hazard ratio (HR) with 95% confidence interval (CI) and *P*-values were calculated to quantify the strength of the associations between the prognostic parameters and survival. Cox proportional hazards model was performed for multivariate analysis. All tests were two sided, and a *P*-value of less than 0.05 was considered to indicate statistical significance.

### Ethics statement

This retrospective study was approved by the Chang Gung Medical Foundation Institutional Review Board (201900611B0). All methods were performed in accordance with the approved guidelines, and written informed consent was waived by the Chang Gung Medical Foundation Institutional Review Board.

## RESULTS

### Patient characteristics

We identified 40 patients with unresectable HCC who received lenvatinib as second-line or later line treatment at our institution between September 2018 and December 2019. There were 35 men and five women with a mean age of 58 (range, 34–78) years. All patients had liver cirrhosis in our study. The Child–Pugh classification was A in 31 (77.5%) patients and B in 9 (22.5%) patients, whereas the BCLC staging classification was B in 3 (7.5%) patients and C in 37 (92.5%) patients. Regarding viral hepatitis, 29 (72.5%) patients had hepatitis B virus (HBV) infection, seven (17.5%) patients had hepatitis C virus (HCV) infection, and four (10.0%) were negative for HBV or HCV. Nineteen (47.5%) patients had macrovascular invasion, including 11 patients (27.5%) with main PVT. In addition, 24 (60.0%) patients underwent hepatectomy before lenvatinib treatment, and extrahepatic spread was noted in 32 (80.0%) patients. There were 17 (42.5%) patients with AFP > 400 ng/mL. At the time of analysis, the median period of follow-up was 15.1 months for the 15 survivors and 8.9 months for all 40 patients. The clinicopathologic parameters of the patients are shown in Table 1.

### Clinical outcomes of lenvatinib treatment

The median PFS of the whole population was 3.3 months (Fig. 1A). Regarding PFS, univariate analysis showed no significant differences in all parameters except AFP > 400 ng/mL. The 17 patients with AFP > 400 ng/mL had worse PFS as compared to the 23 patients with AFP < 400 ng/mL (2.7 months versus 4.2 months, $P = 0.020$, Fig. 2A). Multivariate analysis revealed that AFP < 400 ng/mL ($P = 0.024$; HR, 0.46; 95% CI [0.23–0.90]) was an independent prognostic factor for better PFS.

Regarding OS, the median OS of the whole population was 9.8 months (Fig. 1B). No significant differences were observed in terms of all parameters except AFP > 400 ng/mL in the univariate analysis. As expected, a superior OS was found in 23 patients with AFP <400 ng/mL when compared with the other 17 patients with AFP > 400 ng/mL (not reached versus 6.1 months, $P < 0.001$, Fig. 2B). According to the multivariate analysis, AFP < 400 ng/mL ($P < 0.001$; HR, 0.19; 95% CI [0.08–0.46]) was an independent prognostic factor for superior OS. The results of univariate and multivariate analyses of PFS and OS in 40 patients with unresectable HCC who received lenvatinib after failure of sorafenib treatment are shown in Tables 2 and 3.

### Response and survival

The response to lenvatinib treatment was determined according to the mRECIST criteria, including 1 (2.5%) patient with complete response (CR), 10 (25.0%) with partial response (PR), 16 (40.0%) with stable disease (SD), and 13 (32.5%) with progressive disease (PD). The 6-month PFS rates were 45.7% and 18.8% in the PR and SD groups, respectively; the 1-year OS rates were 53.3% in the PR group, 55.6% in the SD group, and 8.5% in the PD group.

In our study, lenvatinib therapy was used as second-line treatment in 20 (50.0%) patients, third-line treatment in 10 (25.0%) patients, and fourth-line and later lines in 10

**Table 1 Characteristics of 40 patients with unresectable hepatocellular carcinoma who received lenvatinib after failure of sorafenib.**

| Characteristics | |
| --- | --- |
| Age (median) | 58 years (34–78) |
| Body weight | 66.2 kg (44.8–109.5) |
| Liver cirrhosis | 40 (100%) |
| ECOG performance status | |
|     0 | 6 (15.0%) |
|     1 | 34 (85.0%) |
| Sex | |
|     Male | 35 (87.5%) |
|     Female | 5 (12.5%) |
| Child–Pugh classification | |
|     A | 31 (77.5%) |
|     B | 9 (22.5%) |
| BCLC classification | |
|     B | 3 (7.5%) |
|     C | 37 (92.5%) |
| Viral hepatitis status | |
|     Hepatitis B | 29 (72.5%) |
|     Hepatitis C | 7 (17.5%) |
|     No | 4 (10.0%) |
| Main portal vein thrombosis | |
|     Yes | 11 (27.5%) |
|     No | 29 (72.5%) |
| Macrovascular invasion | |
|     Yes | 19 (47.5%) |
|     No | 21 (52.5%) |
| Hepatectomy before lenvatinib | |
|     Yes | 24 (60.0%) |
|     No | 16 (40.0%) |
| Extrahepatic spread | |
|     Yes | 32 (80.0%) |
|     No | 8 (20.0%) |
| AFP > 400 | |
|     Yes | 17 (42.5%) |
|     No | 23 (57.5%) |

**Notes.**

ECOG, Eastern Cooperative Oncology Group; BCLC, Barcelona-Clinic Liver Cancer.

(25.0%) patients. The median PFS and OS were 3.1 months and 8.1 months, respectively, for the second-line treatment; 3.7 months and 11.5 months for the third-line treatment; and 3.1 months and 12.0 months for the fourth line and later line treatments. The results of the survival analyses of the treatment effects of lenvatinib are shown in Table 4.

We also analyzed the correlation of treatment responses between sorafenib and lenvatinib. There were 6 patients with PR to sorafenib previously, and they all achieved

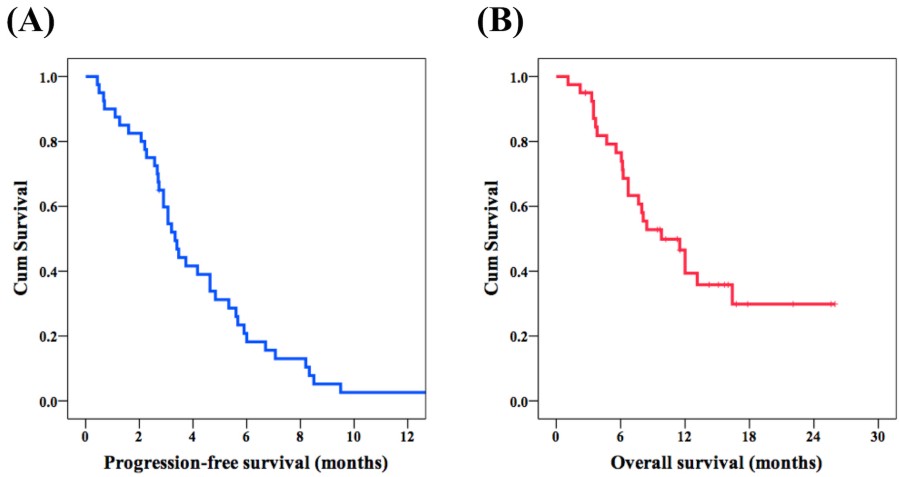

**Figure 1** **Survival outcome of lenvatinib as second-line or later line treatment.** Kaplan–Meier survival curves of progression-free survival (PFS) and overall survival (OS) among patients with unresectable hepatocellular carcinoma who received lenvatinib treatment as second-line or later line treatment. (A) PFS and (B) OS.

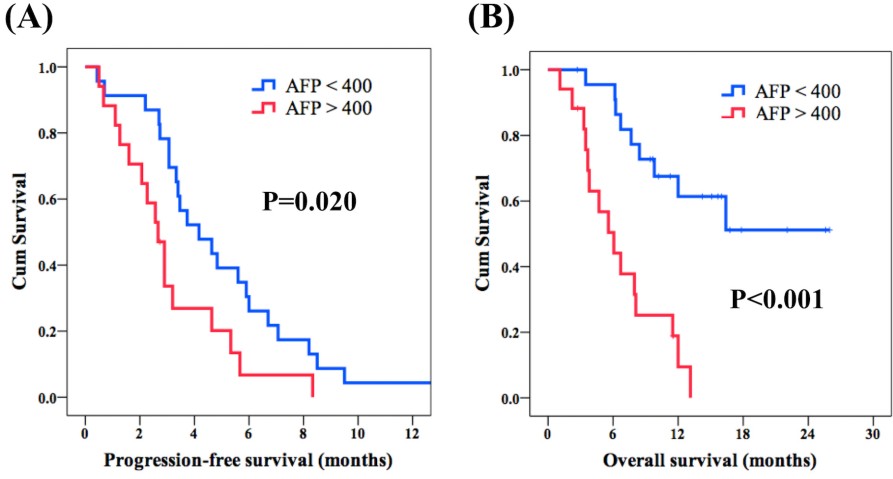

**Figure 2** **Correlation of AFP level and survival outcome.** Comparison of survival curves of progression-free survival (PFS) and overall survival (OS) between patients with unresectable hepatocellular carcinoma who had AFP > 400 ng/mL or AFP < 400 ng/mL. (A) PFS and (B) OS.

CR or PR with lenvatinib treatment. Among 20 patients with SD in response to sorafenib treatment, only 4 (20%) had PR to lenvatinib treatment, and most patients (the other 16 patients, 80%) remained stable with lenvatinib treatment. Among 14 patients who had disease progression with sorafenib treatment, only 1 (7.1%) had PR to lenvatinib treatment, and the other 13 patients showed poor response to lenvatinib treatment. A comparison of treatment response between sorafenib and lenvatinib is presented in Table 5.

**Table 2  Univariate and multivariate analyses of PFS in 40 patients with unresectable hepatocellular carcinoma who received lenvatinib after failure of sorafenib.**

| Characteristics | No. of patients | Univariate | | Multivariate | |
|---|---|---|---|---|---|
| | | Median PFS (months) | P value | HR (95% CI) | P value |
| Age | | | | | |
| <60 years | 22 (55.0%) | 3.2 | 0.40 | | |
| ≥ 60 years | 18 (45.0%) | 3.4 | | | |
| ECOG performance status | | | | | |
| 0 | 6 (15.0%) | 2.7 | 0.70 | | |
| 1 | 34 (85.0%) | 3.3 | | | |
| Sex | | | | | |
| Male | 35 (87.5%) | 3.5 | 0.22 | | |
| Female | 5 (12.5%) | 2.9 | | | |
| Child–Pugh classification | | | | | |
| A | 31 (77.5%) | 3.5 | 0.31 | | |
| B | 9 (22.5%) | 2.9 | | | |
| BCLC staging classification | | | | | |
| B | 3 (7.5%) | 8.2 | 0.22 | | |
| C | 37 (92.5%) | 3.3 | | | |
| Hepatitis B | | | | | |
| Yes | 29 (72.5%) | 5.6 | 0.26 | | |
| No | 11 (27.5%) | 3.2 | | | |
| Hepatitis C | | | | | |
| Yes | 7 (17.5%) | 3.3 | 0.30 | | |
| No | 33 (82.5%) | 6.7 | | | |
| Main portal vein thrombosis | | | | | |
| Yes | 11 (27.5%) | 3.1 | 0.53 | | |
| No | 29 (72.5%) | 3.7 | | | |
| Macrovascular invasion | | | | | |
| Yes | 19 (47.5%) | 2.9 | 0.99 | | |
| No | 21 (52.5%) | 3.5 | | | |
| Hepatectomy before lenvatinib treatment | | | | | |
| Yes | 24 (60.0%) | 3.4 | 0.62 | | |
| No | 16 (40.0%) | 2.9 | | | |
| Extrahepatic spread | | | | | |
| Yes | 32 (80.0%) | 3.1 | 0.27 | | |
| No | 8 (20.0%) | 4.6 | | | |
| AFP level > 400 | | | | | |
| Yes | 17 (42.5%) | 2.7 | 0.020[*] | | |
| No | 23 (57.5%) | 4.2 | | 0.46 (0.23–0.90) | 0.024[*] |

**Notes.**

HR, hazard ratio; CI, confidence interval; ECOG, Eastern Cooperative Oncology Group; PFS, progression-free survival; BCLC, Barcelona Clinic Liver Cancer.
*Statistically significant.

**Table 3** Univariate and multivariate analyses of OS in 40 patients with unresectable hepatocellular carcinoma who received lenvatinib after failure of sorafenib.

| Characteristics | No. of patients | Univariate | | Multivariate | |
|---|---|---|---|---|---|
| | | Median OS (months) | P value | HR (95% CI) | P value |
| Age | | | | | |
| <60 years | 22 (55.0%) | 12.0 | 0.93 | | |
| ≥ 60 years | 18 (45.0%) | 9.8 | | | |
| ECOG performance status | | | | | |
| 0 | 6 (15.0%) | 16.4 | 0.34 | | |
| 1 | 34 (85.0%) | 8.4 | | | |
| Sex | | | | | |
| Male | 35 (87.5%) | 11.5 | 0.48 | | |
| Female | 5 (12.5%) | 3.5 | | | |
| Child–Pugh classification | | | | | |
| A | 31 (77.5%) | 11.5 | 0.99 | | |
| B | 9 (22.5%) | 8.1 | | | |
| BCLC staging classification | | | | | |
| B | 3 (7.5%) | 16.4 | 0.74 | | |
| C | 37 (92.5%) | 9.8 | | | |
| Hepatitis B | | | | | |
| Yes | 29 (72.5%) | NR | 0.23 | | |
| No | 11 (27.5%) | 8.4 | | | |
| Hepatitis C | | | | | |
| Yes | 7 (17.5%) | 9.8 | 0.16 | | |
| No | 33 (82.5%) | NR | | | |
| Main portal vein thrombosis | | | | | |
| Yes | 11 (27.5%) | 12.0 | 0.53 | | |
| No | 29 (72.5%) | 8.1 | | | |
| Macrovascular invasion | | | | | |
| Yes | 19 (47.5%) | 12.0 | 0.13 | | |
| No | 21 (52.5%) | 6.2 | | | |
| Hepatectomy before lenvatinib treatment | | | | | |
| Yes | 24 (60.0%) | 13.1 | 0.09 | | |
| No | 16 (40.0%) | 6.7 | | | |
| Extrahepatic spread | | | | | |
| Yes | 32 (80.0%) | 9.8 | 0.20 | | |
| No | 8 (20.0%) | NR | | | |
| AFP leel >400 | | | | | |
| Yes | 17 (42.5%) | 6.1 | <0.001[*] | | |
| No | 23 (57.5%) | NR | | 0.19 (0.08–0.46) | <0.001[*] |

**Notes.**

HR, hazard ratio; CI, confidence interval; ECOG, Eastern Cooperative Oncology Group; OS, overall survival; BCLC, Barcelona Clinic Liver Cancer; NR, not reach.

*Statistically significant.

**Table 4** Survival analyses of the 40 patients with HCC who received lenvatinib after failure of sorafenib.

| Response to lenvatinib | Number of patients | 6-month PFS rate (%) | P value | 1-year OS rate (%) | P value |
|---|---|---|---|---|---|
| Complete response | 1 (2.5%) | | | | |
| Partial response | 10 (25.0%) | 45.7 | | 53.3 | |
| Stable disease | 16 (40.0%) | 18.8 | 0.046[*] | 55.6 | 0.20 |
| Progressive disease | 13 (32.5%) | 0 | | 8.5 | |

| Lenvatinib treatment lines | Number of patients | PFS (months) | P value | OS (months) | P value |
|---|---|---|---|---|---|
| Second line | 20 (50.0%) | 3.1 | | 8.1 | |
| Third line | 10 (25.0%) | 3.7 | 0.38 | 11.5 | 0.98 |
| Fourth line and later lines | 10 (25.0%) | 3.1 | | 12.0 | |

Notes.
HCC, hepatocellular carcinoma; PFS, progression-free survival; OS, overall survival.
*Statistically significant.

**Table 5** Comparison of treatment response to sorafenib and lenvatinib in the 40 patients with HCC.

| Response to lenvatinib | CR and PR | SD | PD | P value |
|---|---|---|---|---|
| Response to previous sorafenib | | | | |
| PR ($N = 6$) | 6 (100%) | 0 (0%) | 0 (0%) | |
| SD ($N = 20$) | 4 (20%) | 16 (80%) | 0 (0%) | $P < 0.001$[*] |
| PD ($N = 14$) | 1 (7.1%) | 0 (0%) | 13 (92.9%) | |

Notes.
HCC, hepatocellular carcinoma; CR, complete response; PR, partial response; SD, stable disease; PD, progressive disease.
*Statistically significant.

### Adverse events associated with lenvatinib

The AEs of lenvatinib treatment were informed to most patients. AEs with higher frequencies (>20%) included hypertension, diarrhea, decreased appetite, fatigue, and palmar-plantar erythrodysesthesia. Most AEs were grade 1–2; grade 3–4 toxicities were rare, including hypertension (12.5%), diarrhea (2.5%), decreased appetite (2.5%), and fatigue (2.5%). The majority of the patients tolerated the side effects of lenvatinib treatment, and no patients had treatment-related deaths. The profiles of these AEs are shown in Table 6.

## DISCUSSION

HCC is an extremely aggressive malignant cancer, and its treatment is extremely challenging with advanced tumor status. For the past 10 years, sorafenib therapy has been the only recommended first-line systemic treatment for unresectable HCC (*Cheng et al., 2009*; *Llovet et al., 2008*). Although several clinical trials were conducted with several compounds that were designed to be superior or non-inferior to sorafenib, all of them failed, and none were approved (*Cainap et al., 2015*; *Cheng et al., 2013*; *Johnson et al., 2013*). Recently, the REFLECT trial showed that lenvatinib therapy is non-inferior to sorafenib in terms of OS (*Kudo et al., 2018*). Subsequently, lenvatinib was used as the first-line treatment for unresectable HCC in some countries, especially in Japan, and several real-world studies demonstrated that lenvatinib treatment yields good response and has safety (*Hiraoka et al., 2019a*; *Hiraoka et al., 2019b*; *Obi et al., 2019*). However, the clinical impact of lenvatinib

Chen et al. (2020), *PeerJ*, DOI 10.7717/peerj.10382

**Table 6** Treatment-related adverse events of the 40 patients with unresectable hepatocellular carcinoma receiving lenvatinib after failure of sorafenib.

| Adverse event | All patients (N = 40) | | Second line (N = 20) | | Third line (N = 10) | | Fourth and later line (N = 10) | |
|---|---|---|---|---|---|---|---|---|
| | Any grades | Grade 3/4 | Any grades | Grade 3/4 | Any grades | Grade 3/4 | Any grades | Grade 3/4 |
| Hypertension | 19 (47.5%) | 5 (12.5%) | 9 (45.0%) | 3 (15.0%) | 6 (60.0%) | 1 (10.0%) | 4 (40.0%) | 1 (10.0%) |
| Diarrhea | 14 (35.0%) | 1 (2.5%) | 6 (30.0%) | 0 (0%) | 4 (40.0%) | 1 (10.0%) | 4 (40.0%) | 0 (0%) |
| Decreased appetite | 9 (22.5%) | 1 (2.5%) | 4 (20.0%) | 1 (5.0%) | 3 (30.0%) | 0 (0%) | 2 (20.0%) | 0 (0%) |
| Decreased body weight | 7 (17.5%) | 0 (0%) | 4 (20.0%) | 0 (0%) | 1 (10.0%) | 0 (0%) | 2 (20.0%) | 0 (0%) |
| Fatigue | 11 (27.5%) | 1 (2.5%) | 7 (35.0%) | 1 (5.0%) | 2 (20.0%) | 0 (0%) | 2 (20.0%) | 0 (0%) |
| Palmar-plantar erythrodysesthesia | 8 (20.0%) | 0 (0%) | 3 (15.0%) | 0 (0%) | 3 (30.0%) | 0 (0%) | 2 (20.0%) | 0 (0%) |
| Nausea | 6 (15.0%) | 0 (0%) | 3 (15.0%) | 0 (0%) | 2 (20.0%) | 0 (0%) | 1 (10.0%) | 0 (0%) |
| Vomiting | 2 (5.0%) | 0 (0%) | 1 (5.0%) | 0 (0%) | 1 (10.0%) | 0 (0%) | 0 (0%) | 0 (0%) |
| Skin rash | 3 (7.5%) | 0 (0%) | 1 (5.0%) | 0 (0%) | 1 (10.0%) | 0 (0%) | 1 (10.0%) | 0 (0%) |

treatment in patients who received sorafenib treatment remains limited. In our study, the PFS and OS were 3.3 and 9.8 months, respectively, in patients who received lenvatinib after sorafenib; the outcome was similar to that of other second-line treatments for HCC in the previous phase III trials (*Bruix et al., 2017*; *Zhu et al., 2019*). In addition, AFP > 400 ng/mL was noted as an independent poor prognostic factor of PFS and OS in univariate and multivariate analyses.

In the REFLECT trial, patients with Child–Pugh classification B or main PVT were excluded; thus, the efficacy and safety of lenvatinib for these patients are still unclear. However, some patients who had Child–Pugh classification B or main PVT are still treated with lenvatinib in real-world practice. In our study, we enrolled 9 (22.5%) patients with Child–Pugh classification B and 11 (27.5%) patients with main PVT who received lenvatinib therapy as systemic treatment. The median PFS and OS were worse in patients with Child–Pugh classification B or main PVT than in those in patients with Child–Pugh classification A or non-main PVT, although the difference was not significant. This effect was also observed in 19 (47.5%) patients with macrovascular invasion. In contrast, patients who underwent hepatectomy before lenvatinib treatment had better PFS than those who did not, although there was no significant difference. However, the benefit of OS was marginally significant in the hepatectomy group than in the non-hepatectomy group. This could be due to the small size of the patient population in our study.

Sorafenib and lenvatinib have been used in first-line systemic treatment for unresectable HCC, and patients who received lenvatinib as second-line or later line treatment were not enrolled in the REFLECT trial. However, some patients may have received other therapeutic modalities before lenvatinib treatment in clinical practice, such as sorafenib therapy, hepatic arterial infusion chemotherapy, or immunotherapy; thus, lenvatinib therapy is used as second-line or later line treatment rather than first-line treatment. In our study, the median PFS and OS were similar among the second-line, third-line, and fourth-line and later line treatments, although the OS of the second-line treatment was mildly worse than that of the third-line and later line treatments, indicating the efficacy of lenvatinib as second-line and later line treatments for unresectable HCC.

The response rate is also an important issue in HCC treatment. Our study showed that the median PFS was better in responders than in patients with SD or PD, but the 1-year OS rate was equal between patients with PR or SD. In addition, the response to previous sorafenib could predict the response of subsequent lenvatinib. In our study, all patients who responded to sorafenib had treatment response to lenvatinib; most patients (80%) with SD as a response to sorafenib treatment remained in a stable condition with lenvatinib treatment. Patients without response to sorafenib were still not found to be responsive to lenvatinib (92.9%). The correlation between treatment response to sorafenib and lenvatinib is extremely helpful in the treatment of HCC in clinical practice.

Our study had several limitations. First, the study had a retrospective design, with a small sample size, and all patients were treated at a single institution. Second, the number of female patients in our study was limited; therefore, a bias of patient sex might have existed. Third, the follow-up period may not be adequately long, and some survival benefit may not be significant. However, to the best of our knowledge, this is the first study designed

to investigate the clinical impact of lenvatinib as second-line and later line treatment in patients with unresectable HCC who received sorafenib. Further studies with a larger population or a randomized controlled trial are warranted to validate the findings of this study.

## CONCLUSIONS

Our study confirms the efficacy and safety of lenvatinib for use as second-line and later line treatment for patients with unresectable HCC who received sorafenib in clinical practice.

### Funding
The authors received no funding for this work.

### Competing Interests
The authors declare there are no competing interests.

### Author Contributions
- Yen-Yang Chen conceived and designed the experiments, authored or reviewed drafts of the paper, and approved the final draft.
- Chih-Chi Wang and Yueh-Wei Liu performed the experiments, prepared figures and/or tables, and approved the final draft.
- Wei-Feng Li analyzed the data, prepared figures and/or tables, and approved the final draft.
- Yen-Hao Chen conceived and designed the experiments, analyzed the data, authored or reviewed drafts of the paper, and approved the final draft.

### Human Ethics
The following information was supplied relating to ethical approvals (i.e., approving body and any reference numbers):

This retrospective study was approved by the Chang Gung Medical Foundation Institutional Review Board (201900611B0). All methods were performed in accordance with the approved guidelines, and written informed consent was waived by the Chang Gung Medical Foundation Institutional Review Board.

### Data Availability
Raw data is available as a Supplemental File.

### Supplemental Information
Supplemental information for this article can be found online at http://dx.doi.org/10.7717/peerj.10382#supplemental-information.

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
