# Peer review of "Clinical impact of lenvatinib in patients with unresectable hepatocellular carcinoma who received sorafenib"

_PeerJ, doi:10.7717/peerj.10382_

## Round 0.1 · original submission · Major Revisions

Please address all of the reviewers' suggestions because your study was containing limited data. You should clearly explain why your study can impact on hepatocellular carcinoma.

Reviewer 1 ·

Basic reporting

No issues

Experimental design

1) in the abstract and results, it is stated AFP >400 ng/mL is a risk factor: when was this AFP measured? Upon commencement of lenvatinib?

2) In the abstract, methods and results: it is not stated when the mRECIST treatment response is measured- is it 1 month after commencement of lenvatinib? Or maybe 3 months after commencement of lenvatinib?

3) The aims in line 93 are inadequate. They need to be expanded upon, especially with regard to the data about PFS and OS risk factors.

4) In line 99, it is stated "First, only patients who received sorafenib were enrolled." Don't the authors mean "only patients who received sorafenib followed by lenvatinib were enrolled?" Lenvatinib is not mentioned at all in this paragraph!

5) in Line 128, in the statistical analysis, it is stated that "Statistical analyses of group differences were performed using the t-test, Fisher’s exact test, and chi-square test for categorical variables." This is not correct. Firstly, the t-test is a test of continuous, not categorical variables, when means of groups are being compared. Secondly, I see no evidence throughout the rest of the manuscript that the t-test, Fisher’s exact test, and chi-square test are ever applied. The statistical analyses of group differences appear to be performed purely on PFS and OS, for which the log-rank test was used.

6) Similarly, in line 133, it is not stated what test is used to calculate multivariate analysis of the Kaplan-Meier curves. I assume this is Cox proportional hazards testing? But the authors need to make it clear.

Validity of the findings

1) In table 4, there are no statistical analyses (to find p-values) performed between any of the between-group comparisons. Could the authors have done a Chi-square test of 6-month PFS rates, between groups according to response to lenvatinib, with Bonferroni correction? And similarly a Chi-square test of 12-month OS rates, between groups according to response to lenvatinib, with Bonferroni correction? And then log-rank comparisons of PFS and OS between groups according to treatment lines?

2) In table 5, there are no statistical analyses (to find p-values) performed between any of the between-group comparisons. Could the authors have done a Chi-square test, between groups according to response to previous sorafenib, with Bonferroni correction?

·

Basic reporting

1) English language should be implemented. There are some grammatical mistakes especially in the Introduction section

2) In line 114, you wrote that patients >60 kg have been treated with 10 mg of lenvatinib daily. I’m quite sure that this is a typing error, because the correct dose should be 12 mg (based on data from REFLECT trial). If it is not, why did you choose this dose?

3) In line 209, you wrote “Es”, probably instead of “AEs”. This should be corrected

Experimental design

4) In lines 107-108, you wrote that HCC diagnosis was made according to pathology, AFP levels and CT/MRI. Firstly, you should put the reference(s) of guideline(s) you use for HCC diagnosis; secondly, international guidelines have removed AFP levels from diagnosis algorithm, why did you use it?

5) According to international guidelines and phase III trials, second-line treatment after sorafenib should be represented by regorafenib or cabozantinib. Why did you use lenvatinib? In the rationale of the study this is not actually clear

6) Are all 40 patients cirrhotic? This should be reported in population characteristics. If there are non-cirrhotic patients, this characteristic should be added in univariate and multivariate analysis.

7) What type of systemic treatment did patients assume before treatment with lenvatinib in third and fourth-line?

8) The result concerning relationship between the type of response to lenvatinib and type of response to sorafenib is really interesting. It is not clear the reason of discontinuation of sorafenib in the cohort of responder patients (n=26). Have they all developed a radiological progression of disease under sorafenib before initiating lenvatinib? Were there definitive sorafenib discontinuations due to intolerance/toxicity?

9) How did you deal with patients who died before first radiological evaluation of response to lenvatinib? After who many weeks was performed the first CT/MRI of evaluation?

10) AEs incidence is presented for the entire population. Are there significant differences in AEs incidence between second, third and fourth-line cohorts?

Validity of the findings

11) It is not correct making OS analysis on the entire population. Patients receiving lenvatinib as third and fourth-line therapy have been exposed to different drugs that could change tumor molecular characteristics and microenviroment.
It is interesting to note significant differences between second-line cohort OS (8.1 mo) and third/fourth-line cohort OS (11.5 and 12.0 mo, respectively). This could be related to tumor molecular changes due to previous drugs (first of all immunotherapy, as mentioned in the article).

12) As mentioned in Materials and Methods, OS was calculated from the date of lenvatinib initiation to death or last living contact. Probably, lots of patients underwent to a further line therapy after lenvatinib discontinuation. In case of response to the new treatment, we will see an improvement in OS that is not actually related to lenvatinib efficacy. OS data should be presented as general OS and corrected OS by censoring at further line treatment initiation.

Additional comments

Lenvatinib as “non-first-line” therapy for advanced HCC is an interesting field of study. Nowadays, several new drugs are emerging and this study focuses on the future challenge of multistep pharmacological treatment of HCC.
As mentioned in the discussion section, this study has the limitation to be retrospective, monocentric, with a very small sample size and with a short follow-up period. Unfortunately, these are also a number of critical points. First of all the excessive heterogeneity of the population of study. My advertisement is to focus the research on a single cohort of patients (i.e. lenvatinib as second-line after sorafenib? Lenvatinib as third-line after sorafenib and checkpoint inhibitors?) in order to provide data with impact on real clinical practice.

---

## Round 0.2 · accepted · Accept

Dear Chen,

This study showed the efficacy and safety of lenvatinib in patients with unresectable HCC who received sorafenib. It has also shown the safety of lenvatinib as second-line and later line treatment for patients with unresectable HCC who received sorafenib in clinical practice.

Reviewer 1 ·

Basic reporting

No further issues

Experimental design

No further issues

Validity of the findings

No further issues